# The Impact of the COVID-19 Pandemic on the Epidemiology of Influenza in Hospitalised Children in the Years 2017–2025

**DOI:** 10.3390/v18010052

**Published:** 2025-12-30

**Authors:** Zuzanna Wasielewska, Justyna Franczak, Krystyna Dobrowolska, Justyna Moppert, Małgorzata Sobolewska-Pilarczyk, Małgorzata Pawłowska

**Affiliations:** 1Department of Infectious Diseases and Hepatology, Collegium Medicum in Bydgoszcz, Nicolaus Copernicus University in Toruń, 87-100 Toruń, Poland; 2Collegium Medicum, Jan Kochanowski University, 25-369 Kielce, Poland

**Keywords:** influenza, children, COVID-19 pandemic, epidemiology, hospitalisation

## Abstract

Background: The COVID-19 pandemic significantly altered the circulation of respiratory viruses, including influenza. This study aimed to compare the epidemiology and clinical characteristics of paediatric influenza before, during, and after the pandemic. Methods: We retrospectively analysed 553 children aged 0–18 years hospitalised with laboratory-confirmed influenza at a paediatric infectious disease centre in Bydgoszcz, Poland, between September 2017 and August 2025. Patients were stratified into pre-pandemic (A), pandemic (B), and post-pandemic (C) periods. Epidemiological indicators, influenza type, age, sex, and hospital stay duration were assessed using χ^2^ and non-parametric tests. Results: Hospitalisations varied across seasons, lowest in 2021/22 (*n* = 18) and highest in 2024/25 (*n* = 175). Seasonal peaks occurred January–March in groups A and C, whereas group B showed a bimodal pattern in December and March–April. Influenza type A predominated in all periods, though less during the pandemic (56.7% vs. 89.2% pre-pandemic and 73.2% post-pandemic). Median hospital stay decreased from 5 days pre-pandemic to 4 days during and after the pandemic. None of the hospitalised children were vaccinated. Conclusions: The COVID-19 pandemic influenced influenza seasonality, virus type distribution, and hospitalisation patterns in children. Observed shifts highlight the importance of ongoing surveillance and targeted vaccination strategies to mitigate influenza burden in the post-pandemic period.

## 1. Introduction

Influenza is an acute infectious disease caused by influenza viruses of types A and B, with type A accounting for approximately 80% of reported cases worldwide. In recent years, the predominant circulating subtypes have been A/H1N1 and A/H3N2 [1]. Globally, influenza affects approximately one billion individuals annually, with 3–5 million cases progressing to severe disease. In the Northern Hemisphere, influenza activity typically peaks between late autumn and early spring. Transmission occurs primarily via respiratory droplets and aerosols, as well as through direct contact, with an incubation period ranging from 1 to 4 days. High transmission rates are observed in crowded environments such as childcare facilities, schools, healthcare institutions, and long-term care settings.

The clinical presentation of influenza is characterised by fever, headache, fatigue, myalgia, arthralgia, and upper respiratory tract symptoms, including rhinorrhoea, cough, and sore throat. Gastrointestinal manifestations, such as nausea, vomiting, diarrhoea, and abdominal pain, occur less frequently. Certain paediatric populations are at increased risk of severe disease and complications, including children under 5 years of age—particularly those under 2 years—preterm infants, children with chronic cardiac, pulmonary, renal, hepatic, metabolic or haematological disorders (including obesity), oncological patients, and immunocompromised individuals [1,2]. Identification of these high-risk groups is critical when considering antiviral therapy and post-exposure prophylaxis [3].

Common complications of influenza in children include secondary bacterial infections of the respiratory tract (such as pneumonia, otitis media, laryngitis, and pharyngitis), febrile seizures in younger children, and influenza-associated myositis. Diagnostic confirmation is most frequently achieved using rapid antigen detection tests or molecular assays such as real-time polymerase chain reaction (RT-PCR). Although antigen tests are widely available and offer rapid results, their sensitivity and specificity are lower than those of RT-PCR assays, which are generally reserved for hospital settings due to higher costs and technical requirements [2,4]. The availability of multiplex diagnostic tests detecting influenza A and B, SARS-CoV-2, and respiratory syncytial virus (RSV) increased substantially during the COVID-19 pandemic, and since January 2023 such testing has been included among guaranteed services in primary healthcare facilities in Poland.

Antiviral treatment for influenza is available and includes oseltamivir, zanamivir, peramivir, and baloxavir marboxil, with maximal effectiveness achieved when therapy is initiated early after symptom onset [2]. In Poland, oseltamivir remains the most commonly used antiviral agent. The American Academy of Pediatrics recommends oseltamivir due to its favourable safety profile, extensive clinical experience, and ease of administration [3].

Influenza vaccination is recommended for children aged over 6 months and requires annual administration due to the antigenic variability of circulating strains. Vaccine composition is updated annually based on World Health Organization surveillance data [1]. Antiviral treatment is not a substitute for vaccination, and preventive immunisation remains the cornerstone of influenza control [3]. During the COVID-19 pandemic, several international organisations emphasised the importance of maintaining influenza vaccination programmes to reduce the burden on healthcare systems already strained by SARS-CoV-2 infections [5].

In December 2019, cases of pneumonia of unknown aetiology were reported in Wuhan, China, and on 7 January 2020 a novel coronavirus, SARS-CoV-2, was identified as the causative agent of coronavirus disease 2019 (COVID-19) [6]. The rapid global spread of SARS-CoV-2 led to unprecedented public health measures, including lockdowns, school closures, physical distancing, mandatory face coverings, and reorganisation of healthcare services. In Poland, a state of epidemic emergency was declared in March 2020 and remained in force until May 2022 [7,8,9]. On 5 May 2023, the World Health Organization declared that COVID-19 no longer constituted a Public Health Emergency of International Concern [10].

These profound societal and healthcare-related disruptions significantly affected the transmission dynamics of respiratory pathogens. Accumulating evidence suggests that the COVID-19 pandemic not only reduced the circulation of influenza viruses but also altered their seasonal patterns and epidemiological characteristics. However, data describing how these changes translated into paediatric hospitalisation patterns before, during, and after the pandemic remain limited.

Therefore, this study was designed to test the hypothesis that the COVID-19 pandemic was associated with significant modifications in paediatric influenza epidemiology, including changes in seasonality, viral type distribution, and hospitalisation burden. The objective was to compare influenza-related hospitalisations with respect to patient age and sex, influenza virus type, length of hospital stays, and seasonal incidence across pre-pandemic, pandemic, and post-pandemic periods among children hospitalised in a tertiary paediatric infectious disease centre between 2017 and 2025.

## 2. Materials and Methods

### 2.1. Data Collection and Study Population

Data were collected retrospectively from 1 September 2017 to 31 August 2025 at a single tertiary paediatric infectious disease centre in Bydgoszcz, Poland. The study population consisted of 553 children aged 0–18 years hospitalised with laboratory-confirmed influenza, either as a primary or secondary diagnosis, according to the International Classification of Diseases, 10th Revision (ICD-10 codes J10.0, J10.1, and J10.8). Diagnosis was established through positive antigen or molecular (RT-PCR) testing for influenza A or B virus.

Each infectious season was defined as spanning from 1 September of a given year to 31 August of the following year. To examine the impact of the COVID-19 pandemic, patients were stratified into three epidemiological groups:−Pre-pandemic—Group A: September 2017–April 2020.−Pandemic—Group B: September 2021–August 2023.−Post-pandemic—Group C: September 2023–August 2025.

Data from April 2020 to September 2021 were excluded because the hospital functioned primarily as a COVID-19 treatment centre and admitted no patients with other conditions. The period from May to August 2023, when no children were hospitalised for influenza, was pragmatically included in the pandemic-related group (Group B) to maintain continuity across infectious seasons and avoid artificial fragmentation of the pandemic period. This approach allowed for consistent seasonal comparisons and statistical analysis.

The dataset included variables such as patient age, sex, influenza virus type (A or B), and length of hospital stay. Epidemiological data from the District Sanitary and Epidemiological Station in Bydgoszcz were also considered to complement hospital records, as reporting of all confirmed influenza cases is mandatory in Poland.

### 2.2. Ethics

No experimental interventions were administered, and all patients received standard clinical care according to current guidelines [11]. Personal data were anonymised, and informed consent was not required under applicable regulations.

### 2.3. Statistical Analysis

Categorical variables, including age categories, sex, and influenza virus type, were summarised using counts and percentages. Continuous variables, such as length of hospital stay, were reported as medians with interquartile ranges (IQRs) due to non-normal distribution, as assessed by the Shapiro–Wilk test.

Comparisons between groups (A vs. B, A vs. C, B vs. C) were performed using Pearson’s χ^2^ test for categorical variables and the non-parametric Mann–Whitney U test for continuous variables. The Bonferroni correction was applied to account for multiple comparisons.

These pairwise comparisons were pre-specified as the primary analyses to evaluate differences in influenza epidemiology across the three defined periods. Secondary/exploratory analyses included descriptive assessment of monthly and seasonal trends, age group distribution, and potential interactions between influenza type and length of hospitalisation. Statistical significance was defined as a *p*-value < 0.05. Analyses were performed using Statistica v.13 (StatSoft, Tulsa, OK, USA).

## 3. Results

During the study period, 553 children with laboratory-confirmed influenza were hospitalised at the Department of Paediatrics, Infectious Diseases and Hepatology, with an average of 79 admissions per year. The lowest number of hospitalisations occurred in the 2021/22 season, while the highest was recorded in 2024/25 (Table 1).

In all seasons, the lowest number of admissions was observed between June and November. Groups A (pre-pandemic) and C (post-pandemic) demonstrated typical seasonal peaks between January and March. In contrast, Group B (pandemic) displayed a bimodal pattern, with peaks in December and March, and reduced admissions in January and February (Figure 1 and Figure 2).

The mean number of hospitalised patients per season was 74.33 in group A, 60 in group B and 105 in group C. The age of the patients (calculated on the first day of hospitalisation) ranged from 14 days of life to 18 years. No statistically significant differences were observed in age distribution among groups A, B, and C (Table 2).

Figure 3 presents the percentage distribution of the defined age groups of patients across the analysed periods (A, B and C), while Figure 4. shows the number of children in the respective age groups in the analysed infectious seasons.

Sex distribution differed across periods, with a predominance of boys in the pre-pandemic group and more balanced proportions in Groups B and C. The difference between Groups A and C reached statistical significance (*p* = 0.03) (Table 2).

Influenza virus type A predominated across all periods, although its proportion decreased markedly during the pandemic (Group B) compared with the pre- and post-pandemic periods (Groups A and C). Statistically significant differences were observed in all pairwise comparisons (A vs. B, A vs. C, B vs. C; *p* < 0.001) (Table 2, Figure 5 and Figure 6). The 2022/23 season was notable for an equal distribution of influenza types A and B.

The median length of hospital stay decreased from 5 days pre-pandemic to 4 days during and after the pandemic (Groups B and C), with a statistically significant difference between Groups A and C (*p* = 0.03). Longer hospitalisations (≥6 days) were more common in the pre-pandemic period, particularly among children aged ≤2 years, whereas shorter stays (3–4 days) predominated in the post-pandemic period. Figure 7 illustrates the distribution of prolonged hospitalisations (≥6 days) across age groups in the pre-pandemic, pandemic, and post-pandemic periods.

Epidemiological data from the District Sanitary and Epidemiological Station confirmed a marked increase in influenza testing after January 2023, following the introduction of free antigen testing in outpatient clinics. This increase likely contributed to higher numbers of confirmed and reported cases, particularly in 2023–2025 (Table 3). It should be noted that some children admitted to our hospital were from other districts; in these cases, reporting obligations apply to the patient’s district of residence.

Medical history review revealed that none of the hospitalised children had received influenza vaccination during the respective season. This observation reflects persistently low vaccination coverage in the paediatric population rather than vaccine failure.

Overall, the data indicate that the COVID-19 pandemic was associated with altered influenza seasonality, a shift in viral type distribution, shorter hospitalisations, and minor changes in sex distribution among hospitalised children. The temporal patterns observed highlight the impact of pandemic-related public health measures and altered healthcare-seeking behaviour on influenza epidemiology.

## 4. Discussion

Our study is one of the few that compares the epidemiology of influenza in paediatric patients before, during, and after the COVID-19 pandemic. For influenza incidence in the Polish population, the National Institute of Public Health–National Institute of Hygiene–National Research Institute provides data on reported and suspected cases across successive infectious seasons. In the last full season before the COVID-19 pandemic (1 September 2018 to 31 August 2019), 3,954,235 influenza cases were registered in Poland. In the following seasons, the numbers were: 2019/20–3,769,480 cases; 2020/21–1,616,258 cases; 2021/22–2,934,496 cases; and in the 2022/23 season, 5,515,237 cases were reported [12].

Considering the number of children hospitalised with influenza in each infectious season in our analysis, a clear decrease in hospitalisations was observed in the 2021/22 season. It is important to note that during this period, our hospital was designated primarily for the treatment of COVID-19 patients, and routine admissions for other infectious diseases—including influenza—were largely suspended. Therefore, the observed decrease in hospitalisations in our cohort reflects both a true reduction in influenza circulation and the temporary reorganisation of healthcare services rather than a complete absence of influenza cases in the community. In contrast, the 2022/23 season—when most COVID-19-related restrictions in Poland had been lifted—showed an incidence comparable to pre-pandemic levels. Excluding the 2020/21 season, which was omitted due to the specific function of our ward at that time, the national population-level data align with our findings: the fewest children were hospitalised in 2021/22, followed by a marked increase in admissions in the subsequent season. The most recent analysed season, 2024/25, is exceptional—the number of hospitalisations (175) is more than double the average number recorded in pre-pandemic seasons (74 children per season).

When considering influenza type distribution across groups A, B, and C, we observed a marked shift during the pandemic. In groups B and C, influenza type B accounted for a higher proportion of cases compared with pre-pandemic seasons. In particular, the 2022/23 season showed equal proportions of influenza types A and B (50% each), and in 2024/25 influenza type B represented 28% of cases. Prior to the pandemic, influenza type B accounted for less than 6% of cases in most seasons, except for 2017/18 (31%) [13,14]. These findings are consistent with other reports, such as Kondratiuk et al., who identified influenza type B in approximately 1% of Polish children in the 2018/19 season [13]. The shifting dominance of virus types underscores the impact of pandemic-related changes on viral circulation dynamics, including altered exposure patterns and potential viral interference.

In groups A and C, the peak incidence consistently occurred between January and March. However, in group B, we observed a shift in influenza seasonality—the peak incidence occurred earlier (December) and later (March–April) than in groups A and C. This temporal change is likely influenced by both behavioral and healthcare-related factors, including school closures, stay-at-home orders, mask use, reduced social contacts, and changes in healthcare provision, as well as biological or ecological mechanisms, such as viral interference between SARS-CoV-2 and influenza viruses and altered population immunity due to reduced exposure [15,16,17,18]. The pandemic period can thus be conceptualised as a phase of profound systemic disruption, affecting healthcare organisation, public health priorities, and population-level exposure patterns, consistent with observations from Italy and other European countries, where non-pharmaceutical interventions and healthcare system stress reshaped infectious disease circulation at a population level [19]. Figure 2 and Table 1 illustrate these seasonal shifts in hospitalisation numbers and monthly distribution of influenza types.

Analysis of the age of hospitalised patients revealed a decrease in admissions among young children up to 2 years of age and an increase among children over 7 years in group B. In the infectious seasons within group C, children aged 3–6 years represented, for the first time, the smallest proportion of hospitalised patients. Despite these shifts, no statistically significant differences were found between the numbers of children in individual age categories across groups A, B, and C (Table 2, Figure 3 and Figure 4).

The sex distribution of hospitalised children showed the greatest disparity in group A, with boys being hospitalised more frequently than girls (21.98 percentage points). In groups B and C, these differences were substantially smaller: 3.34 percentage points in favour of boys in group B, and 1.9 percentage points in favour of girls in group C. These differences were statistically significant when comparing groups A and C (*p* = 0.03) (Table 2).

The median length of hospital stay was longest in group A (5 days, IQR 3–6), compared with 4 days (IQR 3–5) in groups B and C. Longer stays (≥6 days) affected 32.74% of children in group A, 20.83% in group B, and 18.1% in group C (Figure 7). The shorter hospitalisations observed during and after the pandemic may reflect changes in clinical management, healthcare system pressures, and altered patterns of disease severity.

A particularly notable observation was that none of the hospitalised children had been vaccinated against influenza during the seasons in which they were admitted. This finding should be interpreted with caution, as it primarily reflects persistently low vaccination coverage in the paediatric population in Poland, rather than vaccine failure [19,20,21,22,23,24,25]. According to data from the National Institute of Public Health regarding influenza vaccination, in the years 2017–2019, vaccination coverage in the age groups 0–4 years and 5–14 years did not exceed 1%. A slight increase in the number of vaccinated children has been observed since 2020:−2020: approximately 1.2% in children aged 0–4 years and approximately 2.2% in children aged 5–14 years.−2021: approximately 1.3% in children aged 0–4 years and approximately 1.3% in children aged 5–14 years.−2022: approximately 1.5% in children aged 0–4 years and approximately 1.2% in children aged 5–14 years.−2023: approximately 3.2% in children aged 0–4 years and approximately 1.9% in children aged 5–14 years [20].

Shmueli et al. reported that disruptions to routine childhood immunisation programmes during the COVID-19 pandemic, including reduced access to primary care visits, led to decreased vaccination rates and increased vaccine hesitancy [21]. Influenza vaccination is recommended but not mandatory in Poland, and coverage remains suboptimal despite free vaccination for children aged 6 months to 18 years since September 2021. Internationally, influenza vaccination rates are higher; for example, in the United States, coverage among children aged 6 months to 18 years reached 63.7% in 2019/20 and remained above 57% in subsequent seasons, although a slight downward trend has been observed [22]. Evidence from meta-analyses indicates that influenza vaccination provides substantial protection against hospitalisation, with Boddington et al. reporting 53.2% protection (95% CI: 47.1–58.6) and Kalligeros et al. reporting 57.48% protection (95% CI: 49.46–65.49) [23,24]. The importance of vaccination is further emphasised by Ozsurekci et al., who highlighted that children under 2 years of age constitute a high-risk group for severe influenza, comparable to those with congenital heart defects or oncological conditions [25]. Collectively, these findings underscore the critical need to improve influenza vaccination coverage to reduce hospitalisations and severe disease in children.

Numerous scientific societies and institutions issue recommendations and educational materials regarding influenza vaccination: internationally, WHO, ECDC (European Centre for Disease Prevention and Control), ACIP (Advisory Committee on Immunization Practices, USA) and AAP (American Academy of Pediatrics, USA); and in Poland, the National Institute of Public Health–National Research Institute (NIZP-PZH–PIB), the Polish Paediatric Society and the Polish Vaccinology Society. Across all recommendations, particular emphasis is always placed on the paediatric population [26,27].

Our results highlight the complex interplay between public health interventions, behavioural changes, and biological mechanisms during the COVID-19 pandemic, which collectively influenced the epidemiology of influenza in children. The observed shifts in seasonality, virus type prevalence, age distribution, and hospitalisation patterns illustrate that the pandemic’s impact extends beyond SARS-CoV-2, affecting the circulation of other respiratory viruses. These findings support ongoing surveillance and targeted vaccination strategies to mitigate the burden of influenza in the post-pandemic period.

### Limitations

Several limitations of this study should be acknowledged. First, the monocentric design may limit the generalisability of the results. Second, potential changes in hospital admission criteria over time and the possibility of underdiagnosis of influenza in pre-pandemic seasons may have affected observed trends. Third, data on viral co-infections were not consistently available. Finally, the inclusion of the May–August 2023 period within the pandemic group, while pragmatic to preserve seasonal continuity and avoid artificial fragmentation, may appear arbitrary. Despite these limitations, our study provides valuable insight into temporal trends and factors influencing paediatric influenza during a period of profound epidemiological change.

## 5. Conclusions

The COVID-19 pandemic had a multifaceted impact on the epidemiology of infectious diseases in the paediatric population. Changes in social behaviour, the introduction of mobility restrictions, isolation from educational institutions, and reduced access to healthcare services—including limited preventive visits and disruptions to routine vaccination schedules—affected both the incidence and clinical course of influenza among children. These effects were observed in multiple dimensions, including altered seasonal peaks, shifts in virus type distribution, variations in age and sex patterns among hospitalised children, and changes in the length of hospitalisation.

Based on the data analysed in our study, the observed differences in influenza epidemiology in hospitalised paediatric patients during the COVID-19 pandemic can be linked to a combination of behavioral, healthcare-related, and biological factors. Prior to the pandemic, the predominance of influenza type A was associated with longer hospital stays and was more frequently observed in male patients. During the pandemic, a shift in seasonal patterns was noted, with the typical pre- and post-pandemic peaks in January and February absent, likely reflecting the implementation of stricter sanitary measures, school closures, reduced social contacts, and changes in healthcare provision. In addition, viral interference and reduced population immunity may have contributed to these epidemiological changes.

A notable observation was that none of the children hospitalised in our department during the analysed years had been vaccinated against influenza. This finding highlights persistently low vaccination coverage in the paediatric population in Poland and underscores the importance of promoting influenza vaccination as an effective preventive measure to reduce hospitalisations and severe disease in children.

Further research, including multifactorial analyses of variables influencing the course of influenza and the dynamics of its seasonality, is essential—particularly in the context of the changes triggered by the COVID-19 pandemic and their potential long-term consequences. Continuous monitoring of influenza epidemiology, vaccination coverage, and clinical outcomes in children is necessary to inform public health strategies and to prepare for future seasonal and pandemic influenza threats.

In conclusion, the COVID-19 pandemic represents a phase of profound systemic disruption with long-lasting effects on paediatric infectious disease patterns. Our findings emphasize the need for integrated public health interventions, ongoing surveillance, and improved vaccination coverage to mitigate the burden of influenza in children in the post-pandemic period.

## Figures and Tables

**Figure 1 viruses-18-00052-f001:**
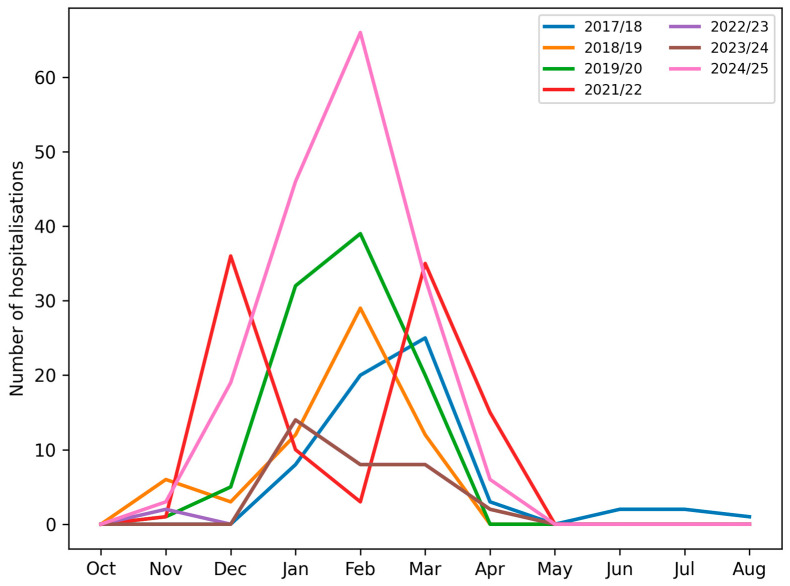
Number of paediatric patients hospitalised with influenza virus type A or type B in consecutive months across subsequent infectious seasons.

**Figure 2 viruses-18-00052-f002:**
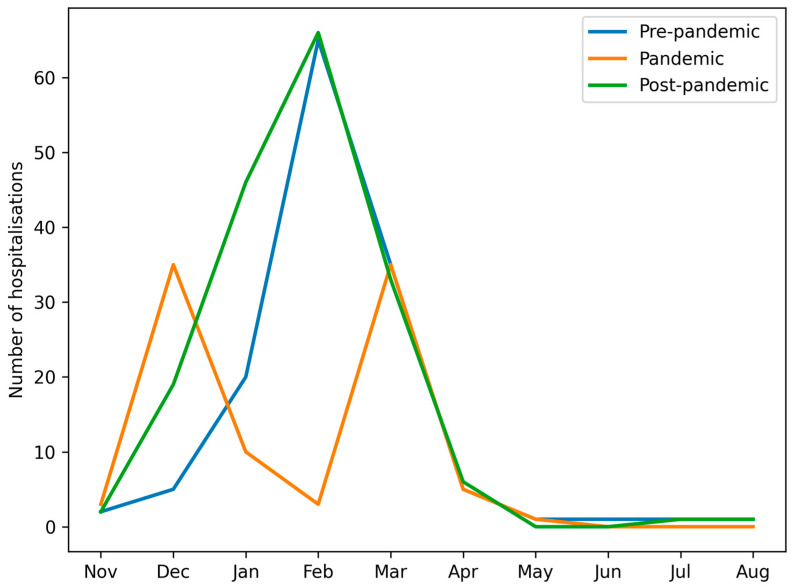
Number of paediatric patients hospitalised with influenza virus type A or type B in consecutive months in groups A, B and C.

**Figure 3 viruses-18-00052-f003:**
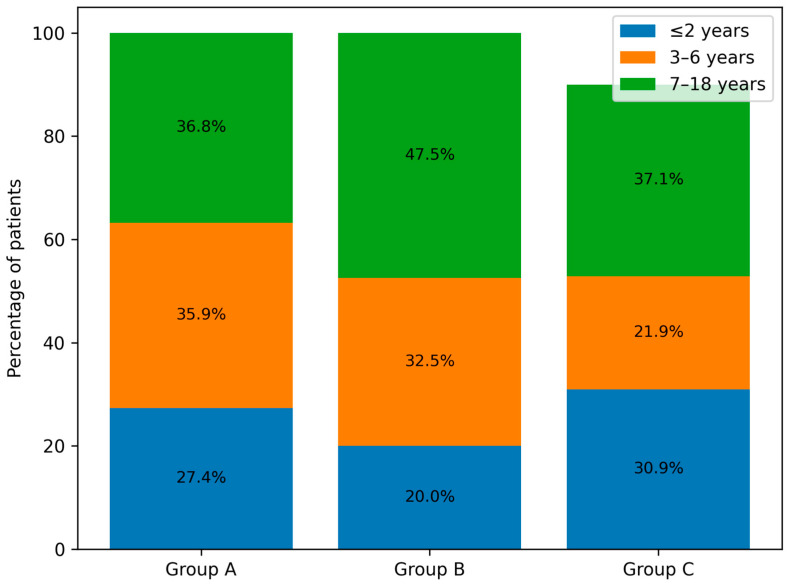
Age distribution of paediatric patients infected with the influenza virus and hospitalised in groups A, B and C.

**Figure 4 viruses-18-00052-f004:**
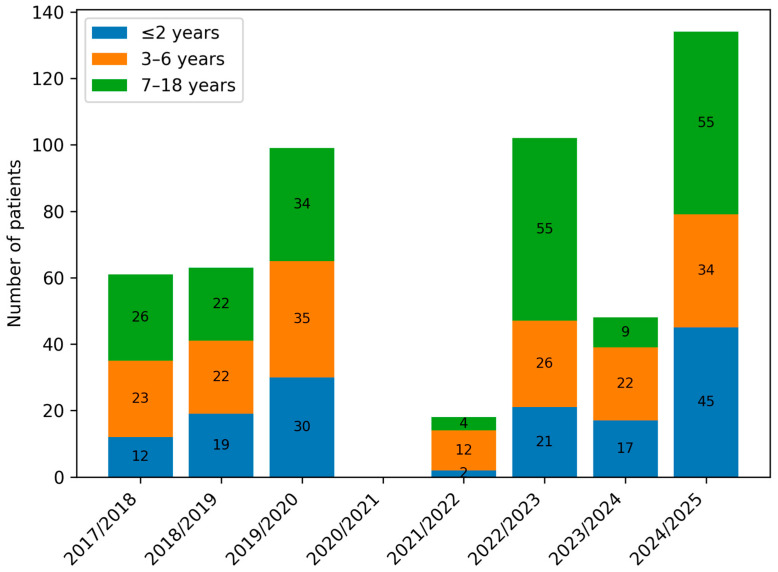
Number of paediatric patients hospitalised with influenza by age group (≤2 years, 3–6 years, and 7–18 years) across consecutive infectious seasons (2017/2018–2024/2025).

**Figure 5 viruses-18-00052-f005:**
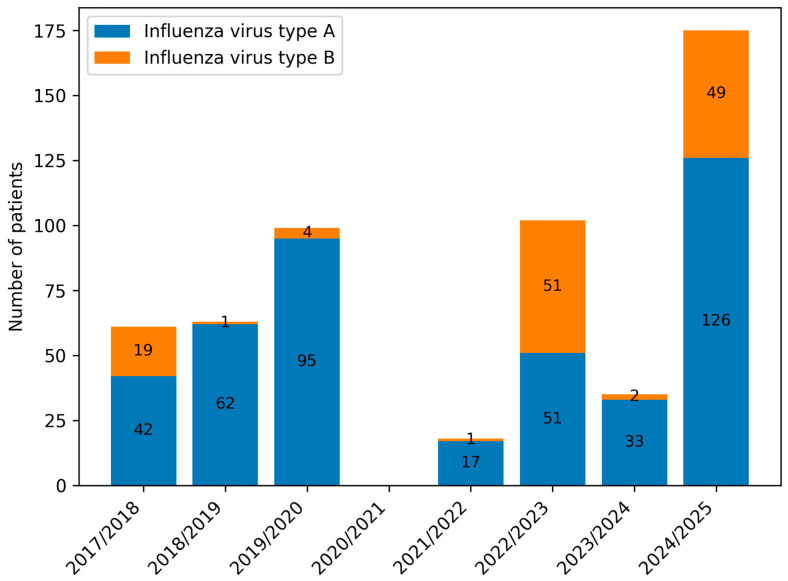
Number of paediatric patients infected with influenza virus types A and B in consecutive infectious seasons.

**Figure 6 viruses-18-00052-f006:**
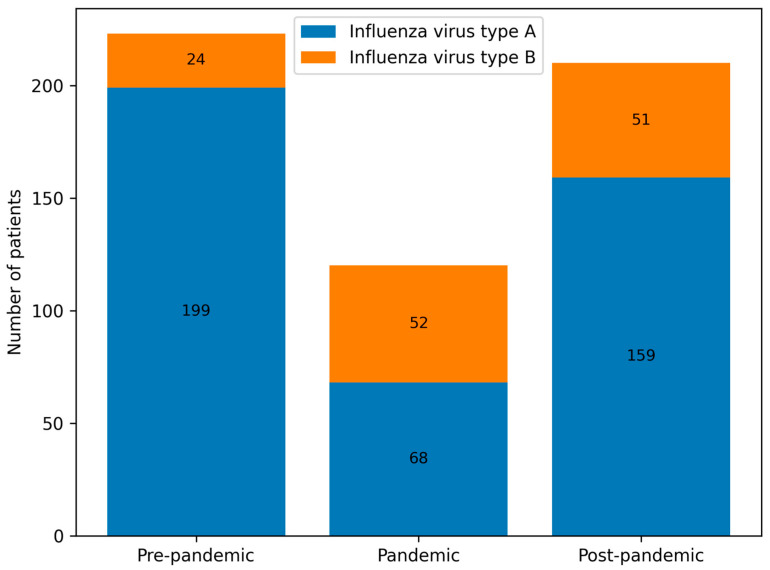
Number of paediatric patients infected with influenza virus types A and B in groups A, B and C.

**Figure 7 viruses-18-00052-f007:**
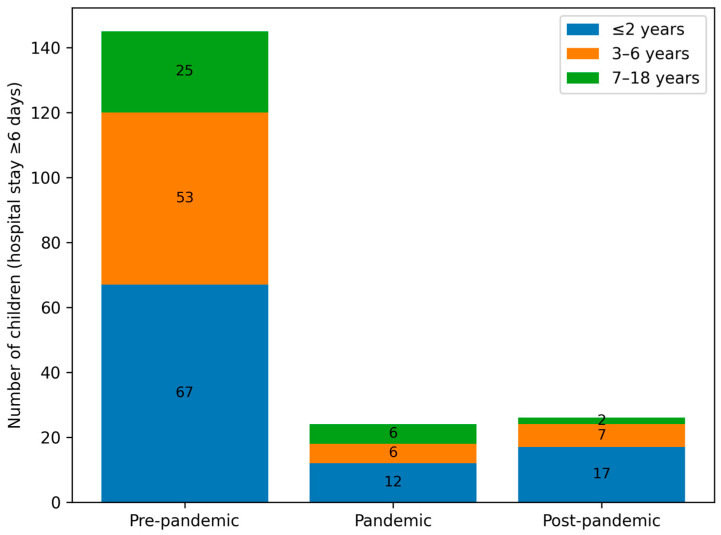
Number of children with prolonged hospitalisation (≥6 days) stratified by epidemiological period (pre-pandemic, pandemic, and post-pandemic) and age group (≤2 years, 3–6 years, and 7–18 years).

**Table 1 viruses-18-00052-t001:** Monthly distribution of paediatric hospitalisations for influenza types A and B, stratified by infectious seasons.

	2017/2018	2018/2019	2019/2020	2020/2021	2021/2022	2022/2023	2023/2024	2024/2025
September	0	0	0	0	0	0	0	0
October	0	1	2	0	0	1	3	2
November	0	6	1	0	1	2	0	3
December	0	3	5	0	0	36	0	19
January	8	12	32	0	0	10	14	46
February	20	29	39	0	0	3	8	66
March	25	12	20	0	3	35	8	33
April	3	0	0	0	6	15	2	6
May	0	0	0	0	7	0	0	0
June	2	0	0	0	0	0	0	0
July	2	0	0	0	1	0	0	0
August	1	0	0	0	0	0	0	0
SUM	61	63	99	0	18	102	35	175
Green—group A			Yellow—group B			Blue—group C		

**Table 2 viruses-18-00052-t002:** Baseline characteristics of paediatric patients infected with the influenza virus and hospitalised in groups A, B and C.

Parameter	09.2017–04.2020 (A) *n* = 223	09.2021–08.2023 (B) *n* = 120	(A) vs. (B) *p*-Value	09.2023–09.2025 (C) *n* = 210	(A) vs. (C) *p*-Value	(B) vs. (C) *p*-Value
Sex, girls/boys, *n* (%)	87 (39.01)/136 (60.99)	58 (48.33)/62 (51.67)	0.3	107 (50.95)/103 (49.05)	0.03	0.99
Age distribution, *n* (%)						
≤2 years old	61 (27.35)	24 (20)	0.36	65 (30.95)	0.99	0.21
3–6 years old	80 (35.87)	38 (32.5)		67 (31.9)		
7–18 years old	82 (36.77)	58 (47.5)		78 (37.14)		
Influenza virus type A/B infection, *n* (%)	199 (89.24)/24 (10.76)	68 (56.67)/52 (43.33)	<0.001	159 (75.71)/51 (24.29)	<0.001	<0.001
Length of hospital stay (days), median (IQR), [min-max]	5 (3–6) [1–21]	4 (3–5) [1–11]	0.06	4 (3–5) [1–13]	0.03	0.99

**Table 3 viruses-18-00052-t003:** Number of positive antigen tests, positive RT-PCR tests, number of hospitalised children and reported influenza cases in the Bydgoszcz district.

Parameter	2017/18	2018/19	2019/20	2021/22	2022/23	2023/24	2024/25
Positive antigen test (*n*)	0	0	0	0	582	466	4575
Positive RT-PCR test (*n*)	14	20	7	9	64	6	29
Number of children hospitalized (*n*)	7	14	4	7	91	28	168
Number of people hospitalized (*n*)	24	40	11	20	337	38	225
Number of ilnesses (adults + children), (*n*)	101	204	62	42	1140	982	8733

## Data Availability

The data obtained for the purposes of the above publication are available from the first author after e-mail contact.

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
