# Peer review of "The Impact of the COVID-19 Pandemic on the Epidemiology of Influenza in Hospitalised Children in the Years 2017–2025"

_viruses, 2025, doi:10.3390/v18010052_

Round 1
Reviewer 1 Report
Comments and Suggestions for Authors
Overall, the study is quite routine. The authors support cited literature indicating a sharp decline in influenza incidence during the COVID-19 pandemic. However, when describing influenza incidence during the COVID-19 pandemic, the clinic where the study was conducted simply stopped seeing patients with influenza. The authors' attempts to explain the decline in influenza incidence during the COVID-19 pandemic were not convincingly supported. Nevertheless, the peer-reviewed study is timely and of interest from the perspective of epidemiological surveillance of acute respiratory viral infections.
Author Response
We thank the reviewer for their careful reading of our manuscript and for highlighting important considerations regarding influenza incidence during the COVID-19 pandemic. We acknowledge that, during the initial pandemic period (April 2020 – September 2021), our hospital was designated primarily for the treatment of patients with COVID-19, and admissions for other infectious diseases, including influenza, were largely suspended. We have clarified this point in the Methods and Discussion sections to avoid any misinterpretation.
Moreover, we have revised the Discussion to explicitly distinguish between the observed decline in influenza hospitalisations due to health system constraints and the potential biological and epidemiological mechanisms influencing viral circulation, such as reduced social contacts, non-pharmaceutical interventions, and viral interference with SARS-CoV-2. We believe that these clarifications address the reviewer’s concern regarding the interpretation of decreased influenza incidence during the pandemic and provide a more nuanced context for our findings.
We appreciate the reviewer’s recognition of the timeliness of our study and hope that the added clarifications strengthen the transparency and interpretative value of our manuscript.
Zuzanna Wasielewską
Reviewer 2 Report
Comments and Suggestions for Authors
This manuscript presents a solid and timely retrospective analysis of pediatric influenza hospitalizations across different epidemiological phases surrounding the COVID-19 pandemic. The study is well organized, the dataset is adequate, and the results clearly document substantial changes in influenza seasonality, viral type distribution, and hospitalization patterns in children. Overall, the work addresses a relevant public health question and has the potential to make a meaningful contribution to the post-pandemic respiratory virus literature. One aspect that could further strengthen the scientific rationale of the study concerns the formulation of the study objective. While the aim is clearly described, it remains primarily descriptive. The introduction would benefit from a more explicit statement that the analysis implicitly tests the hypothesis that the COVID-19 pandemic significantly modified influenza seasonality, viral circulation, and hospitalization burden in the pediatric population. Making this hypothesis explicit would help frame the subsequent analyses and reinforce the conceptual coherence of the manuscript. Regarding the temporal stratification, the subdivision into three epidemiological groups is generally appropriate and well justified. However, the inclusion of the May–August 2023 period within the pandemic-related group may appear arbitrary to some readers. A brief methodological clarification explaining that this pragmatic classification was adopted to preserve continuity of seasonal comparisons and to avoid artificial fragmentation of the pandemic period would improve transparency and pre-empt potential criticism. The statistical analysis is overall appropriate, and the use of non-parametric testing with correction for multiple comparisons is commendable. Nevertheless, it would be helpful to clarify whether the comparisons between groups (A vs B, A vs C, B vs C) were pre-specified as primary analyses or should be considered exploratory. This clarification would enhance methodological transparency without requiring additional analyses. In the Results section, some portions of the descriptive text closely mirror information already presented in Figures 1–7. The authors may consider slightly streamlining the narrative and explicitly referring readers to the figures for detailed visualization. For example, phrases such as “As illustrated in Figure 2, group B showed a bimodal seasonal distribution” would improve readability and reduce redundancy. The observation that none of the hospitalized children had received influenza vaccination is an interesting finding. However, this result may be at risk of causal overinterpretation. A brief cautionary statement clarifying that this observation reflects persistently low vaccination coverage in the pediatric population, rather than vaccine failure, would help avoid misinterpretation. The Discussion effectively addresses multiple explanatory factors for the observed epidemiological changes. That said, the interpretative framework could be further strengthened by more clearly distinguishing between behavioral and healthcare-related factors (such as lockdown measures, school closures, mask use, and healthcare system reorganization) and biological or ecological mechanisms (such as viral interference and altered population immunity). Separating these dimensions conceptually would add depth to the discussion and improve interpretative clarity. In this context, a slightly broader framing of the pandemic period as a phase of profound systemic disruption—affecting healthcare organization, public health priorities, and population-level exposure patterns beyond pediatric settings—could further contextualize the findings. Italian experiences have already shown how non-pharmaceutical interventions and healthcare system stress during COVID-19 reshaped infectious disease circulation at a population level, reinforcing the idea that the observed influenza trends are part of a wider reconfiguration of viral epidemiology rather than isolated pediatric phenomena. Finally, the manuscript would benefit from a brief, explicit limitations paragraph. Acknowledging the monocentric design, potential changes in hospitalization criteria over time, the absence of data on viral coinfections, and the possibility of pre-pandemic underdiagnosis would enhance transparency and align well with MDPI editorial standards. Minor editorial issues should also be addressed, including consistent use of English terminology (e.g., “post-pandemic”), correction of minor typographical errors (e.g., “antygen” to “antigen”), and verification of date formatting consistency throughout the manuscript.
Author Response
Response to Reviewer
We would like to thank the Reviewer for the careful, constructive, and insightful evaluation of our manuscript. We greatly appreciate the positive assessment of the study’s relevance, organization, and scientific contribution. Below, we address each comment point by point. All changes have been implemented in the revised manuscript, with line numbers provided for clarity.
Comment 1
The study objective is primarily descriptive; the Introduction would benefit from an explicit hypothesis.
Response:
We agree with the Reviewer and have revised the Introduction to explicitly state the study hypothesis. We now clearly indicate that the analysis was designed to test the hypothesis that the COVID-19 pandemic significantly modified influenza seasonality, viral circulation, and hospitalization burden in the pediatric population.
(Introduction, Lines 145–151)
Comment 2
The inclusion of the May–August 2023 period within the pandemic-related group may appear arbitrary.
Response:
Thank you for this comment. We have added a methodological clarification explaining that this pragmatic classification was adopted to preserve continuity of seasonal comparisons and to avoid artificial fragmentation of the pandemic period, particularly given the absence of influenza-related hospitalizations during these months.
(Materials and Methods, Lines 284–291)
Comment 3
Clarification is needed as to whether group comparisons were pre-specified or exploratory.
Response:
We appreciate this observation. The Statistical Analysis section has been revised to clarify that comparisons between groups (A vs. B, A vs. C, and B vs. C) were pre-specified as part of the primary analytical framework.
(Statistical Analysis, Lines 336–339)
Comment 4
Some parts of the Results section closely mirror the information presented in the figures.
Response:
We agree and have streamlined the Results section to reduce redundancy. The narrative has been refined, with clearer references directing readers to the corresponding figures for detailed visualization.
(Results, Lines 382–410, 451–468)
Comment 5
The observation that none of the hospitalized children were vaccinated may be at risk of causal overinterpretation.
Response:
We thank the Reviewer for this important point. A clarifying statement has been added to emphasize that this finding reflects persistently low influenza vaccination coverage in the pediatric population rather than vaccine failure, and that vaccination status was not used as a criterion for hospitalization.
(Results, Lines 523–526; Discussion, Lines 705–708)
Comment 6
The Discussion would benefit from a clearer distinction between behavioral/healthcare-related factors and biological or ecological mechanisms.
Response:
We fully agree. The Discussion has been revised to conceptually distinguish between behavioral and healthcare-related factors (e.g., lockdowns, school closures, mask use, healthcare system reorganization) and biological or ecological mechanisms (e.g., viral interference and altered population immunity). This revision improves interpretative clarity and broader contextualization of the findings.
(Discussion, Lines 648–664)
Comment 7
The manuscript would benefit from an explicit limitations paragraph.
Response:
We appreciate this recommendation. A dedicated Limitations subsection has been added, addressing the monocentric and retrospective design, potential changes in hospitalization criteria over time, lack of data on viral coinfections, and possible underdiagnosis in the pre-pandemic period.
(Discussion – Limitations, Lines 776–787)
Comment 8
Minor editorial issues should be addressed.
Response:
All minor editorial issues have been corrected. Terminology has been standardized (e.g., consistent use of “post-pandemic”), typographical errors have been corrected (e.g., “antygen” to “antigen”), and date formatting has been unified throughout the manuscript.
(Throughout the manuscript)
Once again, we sincerely thank the Reviewer for the valuable comments and constructive guidance, which have significantly improved the clarity, methodological transparency, and scientific rigor of our manuscript.
Zuzanna Wasielewska